# PROCRUSTES PROJECTION ALIGNMENT FOR MULTI-VIEW GRAPH REPRESENTATION AND REUSABLE ML MODELS

## ABSTRACT

When a graph is massive or when observability and privacy constraints prevent access to the entire topology, ML models must be trained using only partial information related to the topology. Such models lack reusability when the same graph is specified using a different partial set of measurements or on different subgraphs. We present an approach to make node representations comparable across different graph views produced from the same underlying topology, and use it with Graph Embedding Neural Networks (GENNs) on the OGBN-products benchmark dataset to evaluate its effectiveness. The topology of the graph or a subgraph is captured using the distance to a very small set of anchor nodes, resulting in a view of the graph that depends on the anchors. The dimensionality of these measurements is even further reduced using SVD, and the resulting topology coordinates are used in a GENN scheme. Reusing this model to make predictions on different views of the graph does not produce accurate results. By using a Procrustes transform to align a very small set of reference nodes in views obtained from different sets of anchors, we demonstrate that the models trained on one view can make predictions on the graph based on a different view with about the same accuracy. We also show that the proposed method is accurate when the different views are obtained from different subgraphs with some overlap. The approach requires only a few reference nodes, is compatible with any neural network classifier, and is particularly suitable for privacy-sensitive or federated settings where only projections or a small set of reference nodes can be shared.

## 1 INTRODUCTION

Graph-based learning has become an essential tool for modeling complex relational data in domains such as social networks, recommendation systems, and biological networks. A key challenge in scalable and distributed settings is ensuring that node representations remain consistent and comparable across different graph views and partitions. This becomes particularly important when the complete graph is inaccessible due to privacy constraints, access restrictions, communication and measurement costs, or hardware limitations.

Unsupervised alignment across graphs to enable knowledge transfer when node identities do not overlap is a recurring obstacle in multi-view data integration and cross-graph evaluation (Saxena & Chandra, 2024). Real-world graphs are rarely visible in full to any single analyst because of privacy rules, platforms, access control, time windows, or scale. This limitation means that different people only see partial, view-specific slices (Chiang et al., 2019).

Federated learning demonstrates that privacy rules, regulations, and siloed infrastructure often hinder the centralization of raw data, leading clients to train locally and share model updates while maintaining data decentralization. This setting introduces statistical and system heterogeneity, and non-independent and identically distributed (non-IID) client data can hinder model transfer across parties (McMahan et al., 2017). Classic methods such as FedAvg establish collaborative averaging and communication savings, while FedProx stabilizes optimization under heterogeneity (Li et al., 2020). For graphs, studies and benchmarks report that federated graph learning and federated graph neural networks struggle when clients hold non-IID partitions or structurally different subgraphs

(Xie et al., 2021). Motivated by this setting, we focus on a single underlying graph where the measurements reveal only an anchor-based view of a common node set, i.e., the distance to a node from a small set of anchor nodes. The goal is to reuse a model trained on one view for any other view by performing an orthogonal alignment of embeddings at inference time, thereby preserving locality and enabling cross-site utility.

In this work, we investigate whether node embeddings generated from different anchor-based views of a graph can be made comparable or even interchangeable. If successful, such embeddings would support modular training pipelines, allowing for training on one subgraph and one view and using the same models to evaluate on other subgraphs with different views, or aggregating predictions from multiple views without retraining. We present a method to align multiple views of a graph to achieve this. We design a set of experiments based on the OGBN-Products dataset (Hu et al., 2020) to study this question systematically. Starting with different anchor sets on the complete graph, we explore the consistency of topology-aware node embeddings derived from distance matrices and PCA. We then extend our investigation to settings with partial node coverage by generating subgraphs through node removal. At each step, we evaluate whether embeddings from different views can be aligned using Procrustes analysis (Schönemann, 1966; Even et al., 2024), and whether ML models trained on one view generalize to others. Results presented below indicate that the proposed Procrustes-based projection alignment achieves this without any significant loss of accuracy.

The rest of the paper is organized as follows: the related work in Section 2, introduction of graph coordinates in Section 3, problem statement and our approach in Section 4, our two-phase experiments and results in Section 5, and our conclusion in Section 6.

## 2 RELATED WORK

In recent years, the field of graph representation learning has gained significant attention, especially in methods that aim to accurately map nodes within different graphs for various tasks, including transfer learning across diverse datasets.

Graph researchers have also explored pre-training a model on one graph (or set of graphs) and fine-tuning it on another graph. This paradigm treats the source graph as a pre-training domain to learn general graph feature extractors, which are then adapted to a target graph's node classification task. GraphBridge (Ju et al., 2025) targets arbitrary cross-task and cross-domain transfer with two stages: graph-level pre-training and a tuning stage that bridges mismatched input or output spaces. Our approach focuses on a minimal alignment across views of the same underlying graph rather than learning a transferable backbone across heterogeneous tasks or feature spaces. We compute topology coordinates for each view and estimate a single orthogonal Procrustes map from a small set of unlabeled reference nodes to place all views in a shared coordinate frame, after which a classifier trained on one view can be applied to another without retraining or fine-tuning.

Moreover, the work by Peng et al. (2021) extends the application of Procrustes analysis in knowledge graph embedding (KGE), where embeddings are aligned through closed-form OP methods. Their contributions include a framework that preserves the rich semantics of graphs while facilitating the transfer of learned representations across varying graph structures.

The work by Andreella et al. (2023) lays foundational concepts about utilizing Procrustes methodologies to assess matrix similarity, which can be extended to node embeddings in graph contexts. By leveraging Procrustes distances, researchers can analyze the similarities between embeddings obtained from disparate graphs and enhance performance through effective embedding alignment.

Transferring models trained on one graph to another is increasingly studied. Multi-view and cross-graph methods show that node classification can generalize across related graphs with limited target labels, allowing knowledge from a source graph to improve performance on a target graph. For example, MV-HGNN (Zeng et al., 2024) builds two auxiliary views, a global feature similarity view and a diffusion view, and fuses them with a transformer, sharing information across local, global, and higher-order structures.

A framework for transferring structural information from source domain graphs to target domain graphs by utilizing a pre-training phase is proposed in Wang et al. (2021). A graph neural network is trained using self-supervised learning objectives, which allows it to learn meaningful representations

that can reduce bias when transitioning to the target domain. Their results indicate enhanced performance on recommendation tasks across different domains, providing a clear example of effective model transfer between graphs.

A network tomography problem has been studied by Kakkavas et al. (2021) and Eriksson et al. (2010), they study how to infer internal network properties from end-to-end measurements when the underlying topology is only partially known. Our setting is similar here, as we depend on end-to-end measurements taken by a subset of controllable nodes (anchors) to different nodes in the network. Thus not all the paths in the network are visible. Instead of reconstructing the full topology, we process these distances into anchor-based coordinates and focus on aligning and use them for downstream prediction tasks.

Qin et al. (2023) proposes a coordinate system that utilizes topology coordinates (TCs) as node embeddings. Our work utilizes their embeddings and explores their ability on multiple views of the same graph.

The work by Xu et al. (2019) proposes a method to jointly learn node embeddings and an optimal transport plan that minimizes a Gromov–Wasserstein (GW) discrepancy between the two graphs, solved with a proximal point scheme; alignment and embeddings are coupled, yielding a shared space without an explicit Procrustes step. Compared with our pipeline, GWL optimizes a full GW objective while we estimate a single closed-form rotation on a small reference set and then reuse it to align whole-graph embeddings.

Our work is complementary to recent methods that learn task-specific graph representations and perform cross-graph transfer. Ju et al. (2025) introduces a flexible framework GraphBridge for transferring a pre-trained GNN across heterogeneous tasks and domains by inserting a bridging network that connects input and output layers. The work in Xu et al. (2019) proposes Gromov-Wasserstein Learning (GWL), its scalable variants use optimal transport between metric spaces to jointly match graphs and learn node embeddings. Choudhary & DeCost (2021) proposes a specialized GNN architecture ALIGNN for atomistic systems based on message passing on both the bond graph and its line graph. In contrast, our approach does not learn a new GNN architecture and does not aim to align different graphs. Instead, our technique is aimed at using models trained on one view, in which only distances and a small set of shared nodes are available, to be used with measurements from a different view by aligning multiple local views of the same graph.

## 3 GRAPH COORDINATES

Consider a weighted graph $G$ with $N$ nodes in which $d_{jk}$ is the weighted distance between any two nodes $j$ and $k$. $d_{jk}$ is the lowest sum of weights of the edges between node $j$ and node $k$. Note that for unweighted graphs, $d_{ij} = 1$ for adjacent node pairs, and $d_{ij}$ is the hop distance between $i$ and $j$ for non-adjacent node pairs. Let $\mathbf{D} \in \mathbb{R}^{N \times N}$ be the matrix containing the lowest weighted distance for all two-node pairs, where $N$ is the number of nodes in the graph. This distance matrix can be written as $\mathbf{D} = [d_{ij}]$.

Virtual coordinates (VCs) of a node consist of the vector of distances from a node to a set of $M$ anchor nodes. Without loss of generality, let nodes 1 to $M$ be the set of anchors, i.e., node $N_i$ and $A_i$ are synonymous for $1 \leq i \leq M$. While there are anchor selection schemes in the literature, we randomly select the anchors for simplicity. Thus the VC of $N_i = [d_{A_1 N_i}, d_{A_2 N_i} ... d_{A_M N_i}]$. The matrix $D$ can be reorganized such that its initial $M$ columns and $M$ rows correspond to the selected anchors while maintaining its diagonal elements as zeros. Taking the weighted distances between the $M$ anchors themselves and the weighted distances between the $M$ anchors and the remaining $N - M$ non-anchor nodes, we form the partial distance matrix $\mathbf{P}$, where $j$-th row vector represents the shortest weighted distance from node $j$ to all selected anchors, and is commonly known as the VCs of node $j$.

It is well known that the distance matrix of a graph is low rank, whereas the adjacency matrix, which is the basis of message passing in a GNN, is high rank (Jayasumana et al., 2019). Therefore, a fraction of columns of $D$ can capture the entire topology information (Mahindre et al., 2020).

Topology coordinates of a graph are obtained by the Singular Value Decomposition (SVD) of $\mathbf{P} = \mathbf{U}\mathbf{\Sigma}\mathbf{V}^T$ (Dhanapala & Jayasumana, 2010; 2014). The submatrix consisting of the first $n_c$

columns of $\mathbf{U\Sigma}$, i.e., the most significant $n_c$ principal components, is the topology coordinate (TC) matrix for the graph (Dhanapala & Jayasumana, 2014). The TC of node $N_d$ is given by $C_T(N_d) = [\mathbf{U\Sigma}_{N_d,1}, \mathbf{U\Sigma}_{N_d,2}, ..., \mathbf{U\Sigma}_{N_d,n_c}]$, where $\mathbf{U\Sigma}_{N_d,j}$ is the $N_d$-th row, $j$-th column element in $\mathbf{U\Sigma}$. The calculated TCs are used in different ways as described in the following Section 5.1 and Section 5.2.

When using a small number of anchors, the view is different from the complete graph, because we are not utilizing all edges in the graph. With only 1000 nodes out of 2.45M nodes as anchors, few top principal components already capture nearly all variance in the topology-aware coordinates: the first 10 TCs explain 99.4575% of the energy, leaving only 0.5425% unexplained. Expanding to 50 TCs increases captured energy to 99.6017%, a modest gain of about 0.1442 percentage points, and 100 TCs reach 99.6520%, adding only about 0.0503% more. These percentages show clear diminishing returns beyond a small basis. In practice, 10 to 100 TCs provide an efficient representation that preserves almost all structure; more than 100 TCs will only offer a slight increase that may not translate into measurable improvements in downstream accuracy relative to the added computational cost.

## 4 PROBLEM STATEMENT AND APPROACH

We view the complete graph as a high-dimensional object and each anchor as a camera that senses its distance to different nodes. Thus, the set of anchors produces a limited view of the graph. In multi-view learning, different views share a common latent structure and can be fused or projected into a shared subspace to improve generalization (Xu et al., 2013).

In our method, each anchor set is a projection of the same latent structure; we train on one projection and, after alignment, generalize to others. An analogy from vision: multi-view CNNs for 3D shapes render an object from several viewpoints and aggregate features across views, which strengthens recognition; even a single view is informative when the shared structure is learned (Su et al., 2015). This vision example supports our claim that different viewpoints expose compatible information about the same object. In other words, different anchor sets expose compatible information about the same graph.

In practice, different viewpoints are related by a rigid pose change, such as rotations or flips. In our graph problem, embeddings from different anchor sets of the same graph are related by an approximately orthogonal change of basis. We address this with orthogonal Procrustes: align the embedding from anchor set B to that from A via a rotation. After this alignment, coordinates from different anchor sets become nearly identical, reflecting the shared latent structure emphasized in multi-view learning and the success of cross-view aggregation observed in multi-view CNNs.

Let $G = (V, E)$ be a graph where we only have partial access because of scale. In particular, each view sees the graph through distances to a chosen anchor set rather than by processing the distance to all nodes at once. From different anchor sets $A_i \subseteq V$, we form the view $V_A^i$. Then we construct an embedding for all nodes $V$ in view $V_A^i$: $\Phi_{V_A^i} : V \to \mathbb{R}^k$. Where each row vector is a feature vector derived from distances to anchors, concatenated with attribute vectors, and then reduced to $k$ dimensions. For the same node $j$ in $\Phi_{V_A^i}$ and $\Phi_{V_A^l}$, its embedding $\Phi_{V_A^i}(j)$ and $\Phi_{V_A^l}(j)$ may differ by a rotation or reflection.

We train a classifier $f_\theta : \mathbb{R}^k \to \mathcal{Y}$ on view $V_A^i$ using labeled nodes $V_{train} \subset V$:

$$\theta^\star \in \arg\min_\theta \sum_{j \in V_{train}} \mathcal{L}\Big(f_\theta\big(\Phi_{V_A^i}(j)\big), y_j\Big). \tag{1}$$

Where $\theta^\star$ is the optimal solution and $y_j$ is the label for node $j \in V_{train}$.

To transfer to view $V_A^l$ without retraining, we estimate an orthogonal alignment $R_{V_A^l \to V_A^i}$ using a small reference subset $V_{ref} \subseteq V$ using Procrustes.

Thus, when predicting on $V_A^l$, we have

$$\hat{y}_j = f_{\theta^\star}\big(\Phi_{V_A^l}(j)\, R_{V_A^l \to V_A^i}\big), \quad j \in V \tag{2}$$

Where the model $f_{\theta^\star}$ does not need retraining to predict label $\hat{y}_j$ for node $j$.

Specifically in our method, we treat the embedding from an anchor set A as a matrix $X_A \in \mathbb{R}^{n \times k}$, whose rows $x_A(i)$ are the coordinates of node i and whose columns span a k-dimensional subspace $\mathcal{S}_A = \text{span}(X_A) \subset \mathbb{R}^n$. Let $X_A = U_A R_A$ be a QR factorization with $U_A^\top U_A = I_k$, where $U_A$ is an orthonormal basis of $\mathcal{S}_A$, and $R_A$ is view-specific scaling. Doing the same for another anchor set B, we have $X_B = U_B R_B$. In our setting, both procedures target the same k-dimensional signal subspace $\mathcal{S}$ which is from the complete graph, so $\mathcal{S}_A$ and $\mathcal{S}_B$ are close. Thus, the principal angle between them is small.

We apply Procrustes to topology coordinates, and it solves $\min_{R \in \mathbb{O}(k)} \|X_B - X_A R\|_F = \min_{R \in \mathbb{O}(k)} \|U_B \Sigma_B - U_A \Sigma_A R\|_F$, where $\mathbb{O}(k)$ is an orthogonal group in k dimensions. Because $\Sigma_A$ and $\Sigma_B$ are singular values, and singular values are quite similar for one graph even with different anchor sets, we are actually solving $\min_{R \in \mathbb{O}(k)} \|U_B - U_A R\|_F$. Thus, orthogonal Procrustes can find such R.

In our method, the anchor sets $\{A_i\}$ differ across views. Each anchor set $A_i$ is randomly sampled from $V$, and contains less than 0.05% of the nodes in the graph. There is no special structure required for transfer. The alignment uses a small, randomly chosen $V_{ref} \subseteq V$, which can be entirely unlabeled. In our experiments, $V_{ref}$ also contains less than 0.5% of the total number of nodes in the graph. Allowing overlap $V_{ref} \cap A_i$ is often beneficial: when some reference nodes are also anchors in the first views (the view for training), the shared constraints can benefit the orthogonal transfer and improve prediction accuracy. The classifier $f_{\theta^\star}$ trained on one view is reused for any other view after applying the rotation on the coordinates of the new view, with no further optimization.

## 5 EXPERIMENTS AND RESULTS

We conduct experiments on views of the same underlying graph. We follow the official training/validation/test splits of the OGBN-Products dataset (Hu et al., 2020). The graph is an undirected, unweighted Amazon co-purchase network, has 2,449,029 nodes and 61,859,140 edges. Nodes are products, and edges indicate co-purchases. Each node has a 100-dimensional feature vector obtained by extracting bag-of-words from product descriptions, followed by PCA. The task involves predicting 47-class product categories using accuracy as the evaluation metric. For our views, we keep the nodes and edges fixed and vary only the anchor set used to derive topology features. There is no edge or label changed across views. This isolates the effect of anchor choice while remaining faithful to the OGB split protocol.

We use a lightweight feed-forward network with hidden linear layers and some ReLU activation functions. In our experiments, we have two hidden linear layers with size [128, 64].

We begin with phase 1 to study the simplest controlled setting where the complete graph is available, anchors are randomly chosen, and embeddings from different anchor views can be aligned with Procrustes using a small set of randomly chosen reference nodes. This setting isolates the effect of anchor choice, verifies that views are seeing a nearly common coordinate frame, and establishes baseline transfer when training on one view and evaluating on another without retraining. Phase 2 then relaxes the full coverage assumption by moving to partial views such as subgraphs, which introduces missing nodes and structural variation. The goal is to test whether the alignment procedure and the model trained in partial view remain effective under practical constraints, thereby tracing a clear path from feasibility to robustness.

### 5.1 PHASE 1: FULL GRAPH WITH DIFFERENT ANCHOR SETS

In the initial phase of our experiments on the OGBN-Products dataset, we worked exclusively on the full graph without applying any subgraphing or partitioning. The objective was to evaluate the effect of different anchor sets on the resulting node representations.

We randomly sampled different sets of anchor nodes from the full graph. For each anchor set, we computed the distance matrix from all nodes in the graph to the selected anchors. These distance matrices were then transformed into node embeddings (topology coordinates, TCs) using Principal Component Analysis (PCA). Let $TC \in \mathbb{R}^{N \times d}$ denote the $d$-dimensional TC for $N$ nodes. Coordinates columns are ordered by decreasing explained variance.

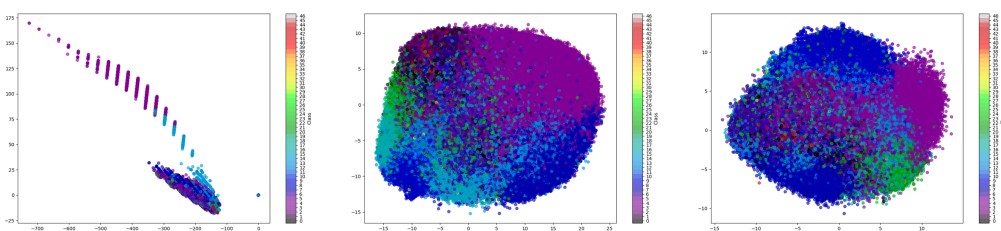

Figure 1: TC plots of anchor set 1 (baseline). Points are nodes, and colors indicate node labels. This view defines the reference coordinate frame used for alignment and model training.

To inspect how geometry and label information unfold across dimensions, we adopt a visualization: for each $i \in \{1, \ldots, k-1\}$ we plot the 2D projection $(\mathrm{TC}_i, \mathrm{TC}_{i+1})$, where $\mathrm{TC}_j$ denotes the $j$-th coordinate column of $TC$. As shown in Fig. 1, we visualized scatter plots of $(\mathrm{TC}_i, \mathrm{TC}_{i+1})$, for $i = 1, 3, 5$ for anchor sets 1 (baseline). Each point is a node; color encodes its class label using a single color map shared across subplots.

To quantify and correct the rotational discrepancy between embeddings, we employed Procrustes analysis. We selected one anchor set as the baseline and used its corresponding TCs as the reference representation.

For each new anchor set, we randomly sampled 1,000 nodes and extracted their TC embeddings. We then applied Procrustes analysis to compute the optimal orthogonal transformation (rotation matrix) that aligns the sampled embeddings from the current anchor set to the reference embeddings from the baseline. This transformation was subsequently applied to the entire set of TCs from the current anchor set to bring it into alignment with the baseline space.

Let $X \in \mathbb{R}^{N \times d}$ denote the matrix of TCs for a set of $N$ nodes from a new anchor set, and let $Y \in \mathbb{R}^{N \times d}$ denote the corresponding TCs of the same nodes derived from a fixed baseline anchor set. The goal of Procrustes analysis (Kabsch, 1976) is to find an orthogonal matrix $R \in \mathbb{R}^{d \times d}$ that best aligns $X$ to $Y$ by minimizing the Frobenius norm:

$$R^* = \arg \min_{R \in \mathbb{R}^{d \times d}, R^\top R = I} |XR - Y|_F^2 \tag{3}$$

This optimization has a closed-form solution. First, compute the cross-covariance matrix $C = X^\top Y$, then take SVD:

$$C = U\Sigma V^\top \tag{4}$$

The optimal rotation matrix is then given by:

$$R^* = UV^\top \tag{5}$$

Once $R^*$ is obtained, we apply it to all topology coordinates $X_{\text{full}}$ of the current anchor set:

$$X_{\text{aligned}} = X_{\text{full}} R^* \tag{6}$$

This transformation brings the embeddings into the coordinate space of the baseline anchor set, enabling direct comparison and cross-anchor evaluation.

We then plot the TCs (before applying Procrustes) of anchor set 2 in Fig. 2, compared to what colors are distributed in Fig. 1, we notice that their shapes are similar, but their orientations are different. As shown in Fig. 3, after applying Procrustes based on the TCs of a small set of reference nodes, even with different anchor sets, the rotated TCs exhibit strong alignment, indicating the effectiveness of the TC and Procrustes transformation. We also conduct an experiment on a third anchor set, where we have similar results. A key observation from this phase was that the PCA embeddings derived from different anchor sets were remarkably consistent, differing only up to an orthogonal transformation (e.g., sign flips or rotations). This invariance suggests a latent structural alignment between representations across different anchor sets, despite the randomness and lack of overlap among the anchor nodes.

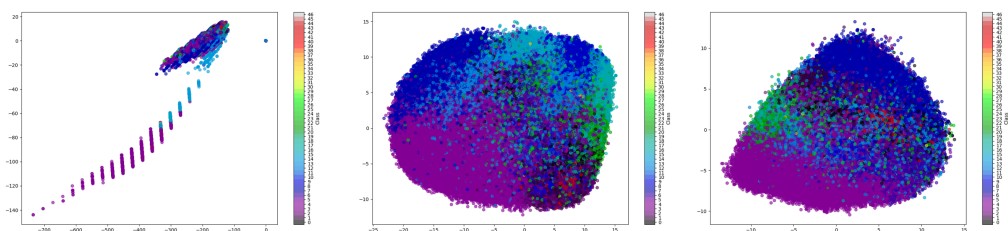

Figure 2: TC plots of anchor set 2, before applying Procrustes to align to the coordinate frame of anchor set 1. A similar global structure is visible, but the mismatch in orientation limits direct reuse of a model trained in the baseline frame.

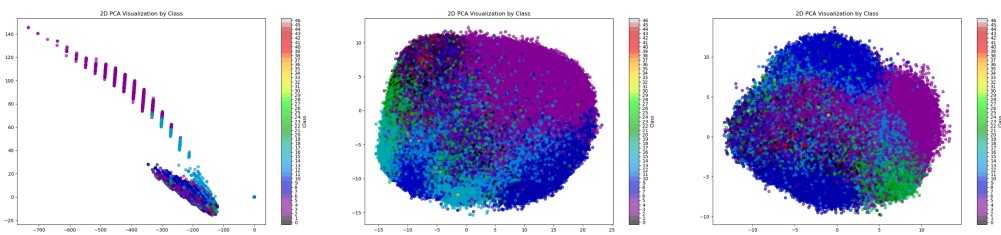

Figure 3: TC plots of anchor set 2, after applying Procrustes to align to the coordinate frame of anchor set 1. The orientation now matches the baseline, enabling training on one and evaluating on another without retraining.

We then conduct an experiment in which a model is trained on one anchor set, and then evaluate this trained model (without retraining or fine-tuning) on other anchor sets. This test aimed to evaluate the generalizability of the learned representations across anchor sets. As shown in Table 1, when trained on anchor Set 1a (baseline), the accuracy on the validation set is 0.8980 and the accuracy on the test set is 0.7714. The same trained model evaluated on anchor Set 2a provides accuracies on validation and test sets of 0.8800 and 0.7599, respectively, which are almost the same as the corresponding values for Set 1a. Similar results can be observed when it is evaluated on anchor Set 3a, where the accuracy on the validation set is 0.8812 and the accuracy on the test set is 0.7580. In comparison, when the proposed Procrustes-based approach is not applied, the validation and test accuracies for anchor Sets 2a and 3a are less than 0.2. These results indicate that a single Procrustes rotation estimated from a small reference is sufficient to place distinct anchor views in a shared coordinate frame, preserving most of the model's predictive power across views.

For TC-based embeddings of the same underlying graph, reference-set Procrustes is a practical, supervision-free step to achieve cross-anchor comparability and reuse trained models.

We also experiment with different numbers of anchors overlapping with anchors from the baseline anchor set. The results are shown in Table 2. This test aimed to evaluate the robustness of the anchor selection across anchor sets. As shown in Table 2, when training on anchor Set 1c (baseline) which has 1000 anchors, the accuracy on the validation set is 0.8977 and the accuracy on the test set is 0.7686; when evaluating on anchor Set 2c with 0% nodes from anchor Set 1c used as anchors, the accuracy on the validation set is 0.8815 and the accuracy on the test set is 0.7521; when evaluating on anchor Set 3c with 50% nodes from anchor Set 1c used as anchors, the accuracy on the validation set is 0.8873 and the accuracy on the test set is 0.7609. The experiments with 100 anchors show similar results. Increasing the percentage of baseline anchors used gives slightly better performance, though the gains are modest. Thus, we can randomly choose anchors when evaluating a trained model on the same graph.

We also explore having some reference nodes overlapping with the anchor in the baseline anchor set. As shown in Table 3, training on the baseline view and evaluating on other anchor views after Procrustes alignment is feasible. For the 1000 anchor setting, aligning from Sets 2e, 3e, 4e to Set 1e yields a small drop from the baseline (validation 0.8977 and test 0.7719) to about 0.876–0.879 on validation and 0.756–0.760 on test, and using overlapping reference anchors provides only modest

Table 1: Train-on-one, evaluate-on-another across anchor sets. The model is trained on anchor set 1 and then evaluated (with no retraining) on other anchor views after Procrustes alignment to the baseline frame.

| Anchor Set | #Anchor | Valid Acc | Test Acc |
|---|---|---|---|
| Anchor Set 1a (baseline, train set) | 1000 | 0.8980 | 0.7714 |
| Anchor Set 2a (TCs NOT aligned to 1a) | 1000 | 0.1237 | 0.1017 |
| **Anchor Set 2a (TCs aligned to 1a)** | 1000 | **0.8800** | **0.7599** |
| Anchor Set 3a (TCs NOT aligned to 1a) | 1000 | 0.1179 | 0.0900 |
| **Anchor Set 3a (TCs aligned to 1a)** | 1000 | **0.8812** | **0.7580** |
| Anchor Set 1b (baseline, train set) | 100 | 0.8702 | 0.7207 |
| **Anchor Set 2b (TCs aligned to 1b)** | 100 | **0.8537** | **0.7125** |
| **Anchor Set 3b (TCs aligned to 1b)** | 100 | **0.8489** | **0.7111** |

Table 2: Train-on-one, evaluate-on-another across anchor sets. The model is trained on anchor set 1 and then evaluated (with no retraining) on other anchor views after Procrustes alignment to the baseline frame, with #Oanc anchors overlapping with anchors in set 1.

| Anchor Set | #Anchor | #Oanc | Valid Acc | Test Acc |
|---|---|---|---|---|
| 1c (baseline, train set) | 1000 | | 0.8977 | 0.7686 |
| **2c (TCs aligned to 1c)** | 1000 | 0 | **0.8815** | **0.7521** |
| **3c (TCs aligned to 1c)** | 1000 | 500 | **0.8873** | **0.7609** |
| 1d (baseline, train set) | 100 | | 0.8702 | 0.7207 |
| **2d (TCs aligned to 1d)** | 100 | 0 | **0.8256** | **0.6813** |
| **3d (TCs aligned to 1d)** | 100 | 50 | **0.8412** | **0.6928** |

gains as #Oref increases from 0 to 1000. In contrast, with only 100 anchors, transfer from Set 1f is noticeably harder: aligned views 2f, 3f, 4f remain well below the baseline 1f on both validation and test, though adding overlapping anchors to the reference set improves validation accuracy somewhat. Overall, alignment supports cross-view transfer; the benefit of overlapping reference anchors is positive but small, and the performance gap is primarily driven by the anchor budget rather than the exact size of the overlapping reference set.

## 5.2 PHASE 2: SUBGRAPH GENERATION BY NODE REMOVAL

In phase 2, we move from the complete graph in phase 1 to subgraphs obtained by removing nodes to mirror the conditions that arise in practice, where coverage of the view is incomplete due to scale, privacy, or distributed storage. Node removal introduces missing vertices and paths, which changes the graph structure and distance-based embeddings, therefore providing a direct stress test for the alignment procedure validated in phase 1. By working with subgraphs formed from random node removal, we model unstructured missingness while keeping the setup simple. With randomly chosen anchors and a small random set of reference nodes for Procrustes alignment, we test whether embeddings from distinct partial views can be placed in a common coordinate frame and whether a model learned under one pattern of missing nodes generalizes to another pattern.

In this phase, we generated subgraphs by randomly removing a subset of nodes from the full OGBN-Products graph. The goal was to simulate scenarios where different parts of the graph are independently accessible, such as in federated or distributed learning settings. Each subgraph retained most of the original structure but had different coverage due to the random node removals.

To get efficiency, we randomly sample a set of nodes (based on a percentage). In our experiments, we randomly remove 30% nodes. After removing these nodes, we extract the remaining largest component and use it as a subgraph.

For each subgraph, we recompute distance matrices based on independently sampled anchor sets and apply PCA to obtain TCs. We then randomly remove 30% nodes from the original OGBN-Products graph to create Subgraph 2. To allow comparison and alignment between these embeddings, we again used Procrustes analysis to align the coordinates of nodes shared between subgraphs. We

Table 3: Train-on-one, evaluate-on-another across anchor sets. The model is trained on anchor set 1 and then evaluated (without retraining) on other anchor views after Procrustes alignment to the baseline frame, with #Anchor nodes as reference nodes, where #Oref reference nodes overlapping with anchors in set 1

| Anchor Set | #Anchor | #Oref | Valid Acc | Test Acc |
|---|---|---|---|---|
| 1e (baseline, train set) | 1000 | | 0.8977 | 0.7719 |
| **2e (TCs aligned to 1e)** | 1000 | 0 | **0.8764** | **0.7560** |
| **3e (TCs aligned to 1e)** | 1000 | 500 | **0.8785** | **0.7580** |
| **4e (TCs aligned to 1e)** | 1000 | 1000 | **0.8786** | **0.7598** |
| 1f (baseline, train set) | 100 | | 0.8702 | 0.7207 |
| **2f (TCs aligned to 1f)** | 100 | 0 | **0.7319** | **0.6077** |
| **3f (TCs aligned to 1f)** | 100 | 50 | **0.7460** | **0.6164** |
| **4f (TCs aligned to 1f)** | 100 | 100 | **0.7523** | **0.6121** |

Table 4: Train-on-one, evaluate-on-another across subgraphs. The model is trained on subgraph 1 and then evaluated (without retraining) on other subgraphs after Procrustes alignment to the baseline frame

| Subgraph | #Anchor | Valid Acc | Test Acc |
|---|---|---|---|
| Subgraph 1a (baseline, train set) | 1000 | 0.8551 | 0.7106 |
| **Subgraph 2a (TCs aligned to 1a)** | 1000 | **0.8387** | **0.6983** |
| **Subgraph 3a (TCs aligned to 1a)** | 1000 | **0.8383** | **0.6975** |
| Subgraph 1b (baseline, train set) | 100 | 0.8362 | 0.6893 |
| **Subgraph 2b (TCs aligned to 1b)** | 100 | **0.8152** | **0.6901** |
| **Subgraph 3b (TCs aligned to 1b)** | 100 | **0.8147** | **0.6888** |

randomly sampled TCs of 10,000 nodes as Procrustes references. When applying Procrustes, we use part of these nodes (if they exist in the new subgraph) as references to derive the rotation matrix.

We also visualize scatter plots of $(TC_i, TC_{i+1})$, for $i = 1, 3, 5$ for Subgraph 1 (baseline), Subgraph 2, and Subgraph 3 as what we do in Phase 1. We have a similar observation from the scatter plots as in Phase 1, where, after applying Procrustes, their shapes are slightly different, but their orientations are similar.

This setup allowed us to investigate further whether the learned representations are transferable not only across anchor sets but also across distinct graph views with partial node overlap.

In our experiments, we randomly remove 30% of the nodes to generate subgraphs, and list the experiment results in Table 4. When training on Subgraph 1a (baseline), the accuracy on the validation and test sets is 0.8551 and 0.7106. When evaluating the trained model on other subgraphs, the valid and test accuracy are 0.8387 and 0.6983 (Subgraph 2a), and 0.8383 and 0.6975 (Subgraph 3a). The results of this phase laid the foundation for more structured partitioning and multi-subgraph training strategies in future work.

These results suggest that using Procrustes as a lightweight, supervision-free commensuration step for train-on-one, evaluate-on-another across subgraphs. These also motivate more structured partitioning (e.g., community splits) and multi-subgraph training in future work.

## 6 CONCLUSION

We addressed a practical reuse problem in graph machine learning: train a node classifier once on a graph view measured with a small number of anchors (using less than 0.05% of nodes as anchors, justification in A.1), such that the same model can be used to make predictions based on other anchor sets and even on other partial views, without retraining. The generalizability of this method is further discussed in A.2. Specifically, we developed a supervision-free route to making node representations comparable across different views of the same graph. We focused on views induced either by distinct anchor sets or by randomly induced subgraphs with partial node overlap. Our key mechanism is an orthogonal Procrustes alignment estimated on a small set of reference nodes; once the rotation is

computed, it is applied to the entire view, producing a shared coordinate frame in which models trained on a baseline view can be used to make predictions on other views and other subgraphs without retraining. The before-and-after Procrustes plots show that, after alignment, both global shape and label color gradients are stable across views, suggesting that the embeddings occupy a common coordinate frame suitable for cross-view transfer.

The proposed method exploits the fact that the distance to an extremely small set of random nodes can capture the graph topology accurately, and in fact, beyond a certain point, additional anchors mainly increase computation cost with only very tiny performance improvement. As a consequence, different independent sets of anchor nodes are capable of capturing the entire topology relationships with high accuracy. By using Procrustes analysis, the proposed approach allows the alignment of these views of the graphs, thereby allowing a ML model based on one view to be used on measurements based on other views. Using TCs without alignment yields validation and test accuracy below 0.2, whereas, after Procrustes alignment, the validation and test accuracies are comparable to the corresponding values of the original baseline view on which the model was trained, which were in the range of 0.7 to 0.9 (Table 1). Our work shows that a single closed-form rotation estimated from a few reference nodes is often sufficient to capture the orientation of views, enabling the reuse of trained models.

The Procrustes solution is determined solely by shared nodes across views. If the reference set is too small or highly localized, the estimated rotation can overfit a local region and not correctly align distant areas. Distributing the reference nodes randomly or increasing the reference set size ensure that reference nodes span the main connected components improves alignment stability.

Our ongoing work involves training on one graph and evaluating the trained model on other graphs without any overlapping nodes. Possible extensions of this work may include transfer learning among different networks corresponding to the same type of data, and also using a common reference frame for multiple entities to train a shared model without disclosing ones own frame of reference.

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

## A  Appendix

### A.1  Justification of Using Low Dimension Principal Components

This section provides further empirical justification for the claim that low-dimensional principal components derived from local graph views are sufficient to capture the global topological structure.

For each dataset and each view, we construct the distance-based matrix used in our topology coordinates, and plot the singular values $\sigma_i$ as a function of the index in Fig. 4. In all plots, we highlight two indices, the smallest index $i$ such that $\sigma_i/\sigma_0 \leq 0.1$ and the smallest index $j$ such that $\sigma_j/\sigma_0 \leq 0.01$.

We repeat this analysis across different anchor set sizes (100 anchors and 1000 anchors), different graphs (OGBN-Products, Cora, CiteSeer datasets).

Across all datasets and views, we observe in the singular value curves that the spectrum decays quickly, with $\sigma_i/\sigma_0$ dropping below $10\%$ and $1\%$ at relatively small indices. This indicates that most of the energy of the distance-based embedding is concentrated in a low-dimensional subspace, and that the leading principal components capture the dominant large-scale topology.

We also observe that, although the number of anchors increases by a factor of ten, the indexes of $10\%$ and $1\%$ remain stable across anchor sizes 100 and 1000. Consequently, low-dimensional principal components derived from local graph views are sufficient to capture the bulk of the topological signal, even when the number of anchors is increased by an order of magnitude.

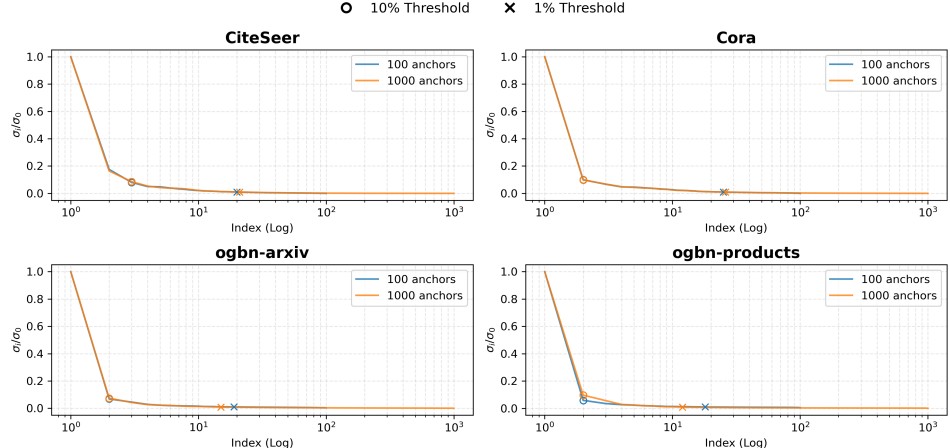

Figure 4: Singular value spectrum of the distance matrix on CiteSeer/Cora/OGBN-Arxiv/OGBN-Products datasets with 100/1000 anchors. The curves show the normalized singular values $\sigma_i/\sigma_0$ as a function of log scale of index. The cross marks the first index where $\sigma_i$ falls below $1\%$ of the largest singular value $\sigma_0$, and the circle marks the first index where $\sigma_i$ falls below $10\%$ of $\sigma_0$.

## A.2 GENERALIZABLITY

Our method depends primarily on topological, rather than node and edge properties of the graph. Our experiment results demonstrate that the alignment requires the following to ensure its generalizability:(1) the shortest path distances that approximate a latent similarity notion, (2) even a very small set of random anchor nodes (less than $0.5\%$ of nodes) are sufficiently distributed across the graph so that their distance vectors form a well-conditioned coordinate system, and similarly, (3) a small set of random reference nodes sufficient to align the different views.

These conditions are not specific to OGBN-Products graph, they hold in other citation networks, social graphs, and knowledge graphs where path distances correlate with semantic similarity. More generally, our alignment scheme is not restricted to distance-based topology coordinates, it can be applied to any graph that has a valid node embedding or coordinate system.

## A.3 THE USE OF LARGE LANGUAGE MODELS (LLMs)

In the process of writing this paper, we used Grammarly, a writing assistant powered by large language models, to enhance the clarity and coherence of my writing.

Grammarly is used to analyze text for grammatical errors and spelling mistakes. The suggestions given by Grammarly are used to fix errors.

Grammarly is also used for stylistic improvements; the review suggestions given by Grammarly are used to improve the description.

