# OpenReview forum: "Procrustes Projection Alignment for Multi-View Graph Representation and Reusable ML Models"
_ICLR.cc/2026/Conference — Submitted to ICLR 2026_

### Official Review · Reviewer_c8Hx · 2025-10-25

**Soundness:** 2
**Presentation:** 2
**Contribution:** 2
**Rating:** 4
**Confidence:** 5

**Summary:**

This paper addresses the challenge of making graph neural network models reusable across different views of the same graph. The authors propose using Procrustes transformation to align node embeddings derived from different anchor-based views, enabling a model trained on one view to make predictions on other views without retraining.  The methodology involves three key steps: (1) computing topology coordinates (TCs) by selecting a small set of anchor nodes (less than 0.05% of total nodes), calculating distances from all nodes to these anchors, and applying SVD for dimensionality reduction; (2) using Procrustes analysis to estimate an orthogonal transformation that aligns embeddings from different views based on a small reference set (less than 0.5% of nodes); and (3) applying this transformation to enable cross-view model transfer. The authors validate their approach on the OGBN-Products dataset through two experimental phases. Phase 1 demonstrates that models trained on one anchor set can achieve comparable accuracy on different anchor sets after Procrustes alignment (test accuracy dropping only from 0.7714 to approximately 0.76 after alignment, compared to 0.10 without alignment). Phase 2 extends this to subgraphs with 30% random node removal, showing the method maintains effectiveness even with partial graph coverage.

**Strengths:**

The paper demonstrates several strengths in its execution and presentation. First, it addresses a practical problem in distributed and privacy-sensitive graph learning scenarios. The motivation is well-articulated: in real-world applications, complete graph access is often impossible due to privacy regulations, access restrictions, or computational constraints. The focus on enabling model reuse across different partial views without requiring centralized data or extensive retraining directly responds to these practical challenges.The experimental design is methodical and well-structured. The two-phase approach provides a logical progression from controlled experiments (Phase 1: complete graph with different anchor sets) to more realistic scenarios (Phase 2: subgraphs with node removal). This progression effectively isolates variables and builds confidence in the approach incrementally. The visualization of topology coordinates before and after Procrustes alignment provides intuitive evidence of the alignment effectiveness, showing how different views can be brought into a common coordinate frame.The computational efficiency of the proposed method is a significant practical advantage. Using only 1,000 anchor nodes and 1,000 reference nodes for alignment represents minimal overhead. The closed-form Procrustes solution requires no iterative optimization, making it substantially faster than methods requiring retraining or complex alignment procedures. The paper effectively demonstrates that 10-100 topology coordinates capture 99.45-99.65% of the variance, showing that the dimensionality reduction is efficient without substantial information loss.The empirical results are convincing within the tested scope. The dramatic difference between aligned and non-aligned embeddings clearly demonstrates the necessity and effectiveness of the Procrustes transformation. The consistency of results across multiple anchor sets suggests the approach is not sensitive to specific anchor choices, which is important for practical deployment.

**Weaknesses:**

The paper's limitation is its lack of fundamental technical novelty. Each component of the pipeline relies on well-established methods:
Topology coordinates are not new to this work. The authors explicitly cite Qin et al. (2023) for the coordinate system and acknowledge it as the foundation of their approach. Computing distance matrices and applying SVD/PCA for dimensionality reduction has been standard practice in graph embedding literature for years. The use of anchor-based distance measurements similarly appears in network coordinate systems and has been explored in the context of sensor networks and graph analysis.
Procrustes analysis for alignment is a classical technique dating back to Schönemann (1966), as the authors cite. The application of Procrustes to graph embeddings is also not novel—the paper cites Peng et al. (2021) who used orthogonal Procrustes for knowledge graph embeddings, and Andreella et al. (2023) for matrix similarity assessment. The related work section reveals multiple prior applications of Procrustes in graph contexts, undermining claims of novelty in the alignment approach itself.
The combination of these techniques, while useful, represents incremental engineering rather than algorithmic innovation. The paper essentially applies a known alignment method to a known embedding approach for a specific use case (anchor-based views). This is valuable engineering work, but it doesn't introduce new mathematical frameworks, theoretical insights, or algorithmic contributions.
Also the paper provides no theoretical analysis of when or why the Procrustes alignment should work.

**Questions:**

1. Under what theoretical conditions does orthogonal Procrustes alignment accurately preserve the geometric structure and predictive information of graph embeddings across different anchor-based views?

2. How does the proposed Procrustes alignment method compare quantitatively against existing cross-graph transfer learning, multi-view graph learning, and federated graph neural network approaches on diverse benchmark datasets?

3. What is the relationship between anchor set size, reference set size, embedding dimensionality, and alignment quality, and how do these parameters affect downstream task performance across different graph scales and structural properties?

4. How robust is the Procrustes alignment approach to realistic graph heterogeneity, including non-overlapping node sets, directed and weighted edges, dynamic graph evolution, varying node/edge feature distributions, and adversarial anchor selection?

**Details Of Ethics Concerns:**

No ethics concerns have been identified in the work.

---

> ### Author Response · Authors · 2025-11-21
>
> **(1) Under what theoretical conditions does orthogonal Procrustes alignment accurately preserve the geometric structure and predictive information of graph embeddings across different anchor-based views?**
>
> Response: We have added an Appendix A.2  to summarize when orthogonal Procrustes alignment preserves the geometric structure and predictive information across anchor-based views. The results demonstrate that  (1) the shortest path distances  approximate a latent similarity notion, (2) even a very small set of random anchor nodes (less than 0.5\% of nodes)  are sufficiently distributed across the graph so that their distance vectors form a well-conditioned coordinate system, and similarly, (3) a small set of random  reference nodes sufficient to align the different views.
>
> When using aligned embeddings for node classification tasks, the class margins are larger than the induced perturbation, so that small changes in the embeddings do not change the argmax of the logits. Thus, under the above conditions, the alignment error is very small in terms of the embedding distortion and the conditioning of the shared nodes, such that orthogonal Procrustes alignment preserve the geometric structure and predictive information.
>
> **(2) How does the proposed Procrustes alignment method compare quantitatively against existing cross-graph transfer learning, multi-view graph learning, and federated graph neural network approaches on diverse benchmark datasets?**
>
> Response: We agree that broader empirical comparisons and multi-dataset evaluations are important. Due to time and computational constraints for this submission, we focused on a detailed, controlled study on OGBN-Products, including extensive configurations over anchors, reference size, and so on. In the final version, we plan to add experiments on additional datasets (like citation and social graphs) and to compare against at least one other alignment baseline to further substantiate generalizability and relative performance. These runs are still in progress and we cannot report stable results within the rebuttal window, but if the paper is accepted we will include the full set of comparative results in the camera-ready version.
>
> **(3) What is the relationship between anchor set size, reference set size, embedding dimensionality, and alignment quality, and how do these parameters affect downstream task performance across different graph scales and structural properties?**
>
> Response: We agree that the relationship between anchor set size, reference set size, embedding dimensionality, and alignment quality is important for understanding and applying our method. We now discuss these dependencies explicitly in the Conclusion section.
>
> Results presented demonstrate that even when using less than 0.5\%  of randomly selected nodes for anchors, components, or reference nodes, they are sufficiently  well-spread to recover a globally consistent alignment. Increasing the anchor, component, or reference set size beyond a certain point  brings only marginal gains.
>
> We are currently running more sets of experiments that sweeps over these hyperparameters, we expect to include the corresponding numerical results in the final version.
>
> **(4) How robust is the Procrustes alignment approach to realistic graph heterogeneity, including non-overlapping node sets, directed and weighted edges, dynamic graph evolution, varying node/edge feature distributions, and adversarial anchor selection?**
>
> Response: Our experiments already vary the overlap between views and allow anchor and reference sets to be non-overlapping. The more extreme case where two views have disjoint node sets (such as partitioning the graph into two independent components) is outside our current scope: if global topology is split, an orthogonal transform cannot recover information that is simply absent. However, if two graphs are structurally similar (such as different snapshots or related networks), it is an interesting open question whether the same scheme can still succeed, and we plan to explore this in future work.

---

### Official Review · Reviewer_imen · 2025-10-28

**Soundness:** 2
**Presentation:** 2
**Contribution:** 3
**Rating:** 4
**Confidence:** 3

**Summary:**

This paper addresses the problem of cross-view model reuse in graph machine learning, where node classifiers trained on one graph view (defined by a specific anchor set or subgraph) should be applicable to other views without retraining. The authors propose a Procrustes-based orthogonal alignment method to make node embeddings from different views comparable. Using topology coordinates derived from anchor-node distances, they demonstrate that a simple rotation transformation estimated from a small reference node set can effectively align embeddings across views. Experiments on OGBN-Products show that models trained on one view maintain accuracy when applied to aligned embeddings from other views, whereas direct cross-view application fails completely.

**Strengths:**

(1) The paper creatively applies Procrustes analysis to align graph embeddings from different anchor-based views, a setting not extensively studied in prior work.

(2) The approach is lightweight, requires no labels for alignment, and has direct relevance to federated and distributed graph learning.

(3) The authors show that their method works effectively in both full-graph and subgraph settings, with significant gains over the no-alignment baseline.

**Weaknesses:**

(1) The paper only compares against a "no alignment" baseline. It does not evaluate against other embedding methods or alignment techniques, making it difficult to assess the relative merit of the proposed approach.

(2) Experiments are conducted only on one dataset (OGBN-Products) and one task (node classification). Broader evaluation across datasets and tasks is needed to establish generalizability.

(3) The paper relies entirely on empirical validation and lacks theoretical analysis.

**Questions:**

(1) Can you add comparative experiments against state-of-the-art graph embedding and alignment methods to better demonstrate the relative effectiveness of your approach?

(2) Could you evaluate the method on more diverse datasets and tasks to further validate its generalizability?

(3) Could you provide a theoretical analysis?

---

> ### Author Response · Authors · 2025-11-21
>
> **(1) Can you add comparative experiments against state-of-the-art graph embedding and alignment methods to better demonstrate the relative effectiveness of your approach?**
>
> Response: We agree that additional comparisons would better demonstrate the effectiveness of our approach. During the rebuttal period we have started running new experiments: (1) compare against representative state-of-the-art graph embedding/alignment baselines, (2) apply all methods on subgraphs constructed purely from shortest path edges, and (3) extend our evaluation to additional datasets, including citation benchmarks (Cora, PubMed datasets) and ogbn-proteins. Due to time and computational constraints, these runs are still in progress and we cannot report stable results within the rebuttal window, but if the paper is accepted we will include the full set of comparative results in the camera-ready version.
>
> **(2) Could you evaluate the method on more diverse datasets and tasks to further validate its generalizability?**
>
> Response: We added a section in Appendix A.2 to discuss its generalizability. The method related  primarily to the graph’s topological structure rather than its domain-specific node or edge attributes.
> We are currently running more experiments on diverse datasets (Cora, CiteSeer, PubMed). Due to time and computational constraints, these runs are still in progress.
> We plan to include the full set of comparative results in the final camera-ready version.
>
> Furthermore, in Related Work, we have added a paragraph related to network tomography, which exactly corresponds to the problem addressed in this paper. Specifically, in many tomography scenarios, there is a small set of nodes (akin to our anchor nodes) that are capable of making measurements, e.g., distance, roundtrip time, etc., to other nodes in the network.  Our method allows, e.g., different vendors with each having their own measurement nodes and thus different network views, to collaborate at ML model level applications.
>
> **(3) Could you provide a theoretical analysis?**
>
> Response: We agree with the reviewer that a theoretical analysis of the proposed alignment is important. However, given the length constraints, our focus in this submission  was on the  methodology and empirical behavior.  A theoretical analysis will be presented as well as a more general context in a follow up publication.

---

### Official Review · Reviewer_FGTR · 2025-10-28

**Soundness:** 2
**Presentation:** 3
**Contribution:** 2
**Rating:** 4
**Confidence:** 3

**Summary:**

This paper proposes a method for aligning node representations across different anchor-based graph views or subgraphs derived from the same underlying topology. The authors employ distance-based topology coordinates (TCs) combined with orthogonal Procrustes alignment to enable model reuse and cross-view generalization under settings where graph observability is limited due to privacy or scale. The work is motivated by practical challenges and is clearly presented. However, the paper lacks theoretical depth, rigorous novelty justification, and analytical evaluation.

**Strengths:**

(1) The paper addresses a realistic and timely issue: training graph models under partial observability or privacy constraints, and reusing them across different subgraph views.
(2) The proposed framework (distance matrix + PCA + Procrustes) is simple, transparent, and easy to replicate, which facilitates interpretability and reproducibility.

**Weaknesses:**

(1) Orthogonal Procrustes alignment and cross-view embedding matching have been well studied in previous work (e.g., Peng et al., 2021; Andreella et al., 2023). The proposed method applies existing ideas in a new context but lacks a substantial methodological innovation.
(2) The paper provides no error bounds, stability guarantees, or theoretical justification for the effectiveness of Procrustes alignment in this context. It remains unclear why low-dimensional principal components derived from partial distance matrices can sufficiently capture global topological structure.
(3) Experiments rely solely on the OGBN-Products dataset. The absence of diverse benchmarks (e.g., citation, social, or knowledge graphs) undermines claims of generalizability. No baseline comparisons are provided with established methods for graph alignment or transfer learning.

**Questions:**

(1) Could the authors offer further theoretical or empirical justification clarifying why the low-dimensional principal components obtained from local graph views are sufficient to capture and represent the global topological structure?
(2) Do the authors expect their approach to generalize to graphs of different domains or scales (e.g., citation networks, social graphs, or knowledge graphs)? If so, what properties of the proposed method ensure this generalizability?
(3) Could the authors comment on how their approach compares against other graph alignment or transfer-learning methods such as GraphBridge, GWL, or ALIGNN? Even a qualitative or computational comparison would help contextualize the claimed advantages.

---

> ### Author Response · Authors · 2025-11-21
>
> **(1) Could the authors offer further theoretical or empirical justification clarifying why the low-dimensional principal components obtained from local graph views are sufficient to capture and represent the global topological structure?**
>
> Response: We thank the reviewer for raising this important point. Our approach relies on the fact that distances to a small, not tightly clustered set of anchors provide an coordinate system that already encodes long-range connectivity patterns. In the revision submission, we add the justification in Appendix A.1, we report the proportion of variance captured by the top
> $k$ principal components for each local view with multiple anchor sizes, showing that a small number of components capture the vast majority of the variance.
>
> **(2) Do the authors expect their approach to generalize to graphs of different domains or scales (e.g., citation networks, social graphs, or knowledge graphs)? If so, what properties of the proposed method ensure this generalizability?**
>
> Response: Yes, we do expect our approach to generalize across different graph domains and scales. We have added the generalizability justification in Appendix A.2. The method depends primarily on the graph’s topological structure rather than on domain-specific node or edge attributes.
> The results demonstrate that  (1) the shortest path distances  approximate a latent similarity notion, (2) even a very small set of random anchor nodes (less than 0.5\% of nodes)  are sufficiently distributed across the graph so that their distance vectors form a well-conditioned coordinate system, and similarly, (3) a small set of random  reference nodes sufficient to align the different views.  These conditions have been shown to be generally applicable to graphs associated with many other datasets associated with real-world applications,  e.g., see: Ref [Inference in Social Networks from Ultra-Sparse Distance Measurements via Pretrained Hadamard Autoencoders, DOI: 10.1109/LCN48667.2020.9314769] and Ref [Graph Coordinates and Conventional Neural Networks - An Alternative for Graph Neural Networks, DOI: 10.1109/BigData59044.2023.10386792]. As  a result, the method can be expected to generalize across many domains.
>
> Furthermore, as the response to Question 1 of Reviewer 2 below indicates, the final camera ready version will contain  results demonstrating the generalizability over several additional datasets.
>
> **(3) Could the authors comment on how their approach compares against other graph alignment or transfer-learning methods such as GraphBridge, GWL, or ALIGNN? Even a qualitative or computational comparison would help contextualize the claimed advantages.**
>
> Response: We thank the reviewer for this suggestion. Our approach targets a different setting from GraphBridge, GWL, and ALIGNN, and we now make this clearer in the Related Work section.
>
> GraphBridge and related transfer-learning frameworks are designed to adapt a pre-trained GNN across different source/target graphs and tasks, typically by adding learnable bridging modules and fine-tuning the network. In contrast, our method stays within a single large graph and focuses on aligning multiple anchor-based local views via PCA and closed-form Procrustes alignment, without training an additional deep model on the target graph.
>
> GWL optimizes a full GW objective while we estimate a single closed-form rotation on a small reference set and then reuse it to align whole-graph embeddings.
>
> ALIGNN is a domain-specific architecture for atomistic graphs that enriches message passing using line graphs. In contrast, our approach does not learn a new GNN architecture and does not aim to align different graphs. Instead, we study whether classical topology-based embeddings and orthogonal Procrustes alignment are sufficient to align multiple local views of the same graph in a scenario where only distances and a small set of shared nodes are available.

---

### Meta-Review · Area_Chair_TkLt · 2026-01-05

**Summary:**

This paper studies the problem of model reuse across different anchor-based views or subgraphs of the same large graph under partial observability and privacy constraints. The authors propose a simple pipeline based on distance-based topology coordinates, PCA/SVD, and orthogonal Procrustes alignment to align embeddings across views, enabling a classifier trained on one view to generalize to others without retraining. Experiments focus primarily on OGBN-Products and include both full-graph and subgraph settings.  Reviewers broadly agree that the problem is realistic and practically motivated, and that the method is simple, interpretable, and computationally efficient. However, they also consistently raise concerns regarding limited methodological novelty, lack of rigorous theoretical guarantees, restricted experimental scope, and insufficient comparisons to existing graph alignment and transfer methods. While the rebuttal provides clarifications and additional justification in appendices, several core concerns remain only partially addressed.

**Reviewer Concerns:**

Concerns addressed by the rebuttal (partially):

The authors provided additional empirical justification for the use of low-dimensional principal components and discussed generalizability conditions in appendices.

The scope of the method relative to GraphBridge, GWL, and ALIGNN was clarified in the revised related-work discussion.

The authors articulated informal conditions under which Procrustes alignment preserves predictive information.

Other concerns:

Limited novelty: The core components (anchor-based distance embeddings, PCA, Procrustes alignment) are well-established. The contribution is largely an application and integration of known techniques rather than a new algorithmic or theoretical advance.

Lack of theory: No formal error bounds, stability guarantees, or alignment guarantees are provided. The rebuttal offers intuition but does not supply rigorous analysis.

Narrow empirical scope: Experiments remain centered on a single dataset (OGBN-Products) and one task (node classification). Additional datasets and baseline comparisons are deferred to a hypothetical camera-ready version.

Weak comparative evaluation: The paper compares primarily against a “no-alignment” baseline. Quantitative comparisons with existing graph alignment, transfer, or federated graph learning methods are missing.

**Reviewer Scores:**

Taking into account the rebuttal but also its limitations. I don't think the scores will change.

---

### Decision · Program_Chairs · 2026-01-26

Reject